# High-temperature thin-film thermocouple for aero-engines

**Keying Guo[1], Yazhong Lu[2]\*, Zhihui Liu[3]**

**1** Zhejiang University-University of Illinois Urbana-Champaign Institute, Zhejiang University, Hangzhou, China, **2** School of Mechanical Engineering, Zhejiang University, Hangzhou, China, **3** State Key Laboratory of Tribology in Advanced Equipment, Tsinghua University, Beijing, China

\* luyazhong@zju.edu.cn

## Abstract

A thin-film thermocouple sensor directly deposited on nickel-based superalloys was investigated for aero-engine applications. To address the high-temperature and harsh environment challenges, the sensor's film structure was meticulously designed. A NiCrAlY thin-film transition layer was deposited via magnetron sputtering, while an $Al_2O_3$ thin-film insulation layer and protective layer were prepared using electron beam evaporation and RF magnetron sputtering. Additionally, a Pt10Rh/Pt thin-film sensitive layer was fabricated via DC magnetron sputtering. Results indicate that the thin-film thermocouple sensor can operate at temperatures up to 1100 °C, with a Seebeck coefficient of 8.16 μV/°C, a maximum temperature error below ±1%, and a service life exceeding 20 hours. This sensor withstands the harsh conditions of aero engines, significantly improves local temperature measurement accuracy, and offers valuable insights for turbine blade lifespan and cooling design.

## 1. Introduction

Modern aero-engines operate at turbine inlet temperatures exceeding 1600°C to achieve higher thrust-to-weight ratios. However, such extreme conditions subject turbine blades to complex thermal-mechanical loads, leading to oxidation, creep, and fatigue failures. Precise temperature measurement is critical to ensure operational safety, optimize cooling efficiency, and extend component lifespans. Traditional thermocouples, while widely used, suffer from limitations such as intrusive installation, slow response (>1 s), and spatial resolution constraints, which disrupt airflow and fail to capture localized thermal gradients. Thin-film thermocouples (TFTCs), with their ultra-thin profile (<10 μm) and direct deposition capability, offer a transformative solution. By integrating seamlessly onto blade surfaces, TFTCs enable real-time monitoring of transient thermal loads without perturbing the aerodynamic profile. Despite advancements in material selection (e.g., Pt-Rh, W-Re) and flexible designs, challenges persist in achieving robust adhesion, high-temperature insulation,

**Data availability statement:** All data are in the manuscript.

**Funding:** The author(s) received no specific funding for this work.

**Competing interests:** The authors have declared that no competing interests exist.

and long-term stability under cyclic thermal and mechanical stresses. This study addresses these gaps through a novel multi-layer TFTC structure.

Recent research has focused extensively on the development and optimization of thin-film thermocouples. Studies have explored improvements in sensitivity, stability, and integration for high-temperature environments. For instance, Wang et al. [1] and Zhang et al. [2] investigated online emissivity-based methods and high-temperature sputtering processes, respectively. Ma et al. [3] and Zhang et al. [4] examined material properties and thermal insulation mechanisms for enhanced durability. Additionally, Ruan et al. [5] developed high-precision thermocouple systems, while Tianbian et al. [6] addressed multi-node sensor array structures for spatial temperature distribution. Research has also expanded into innovative material and fabrication techniques. Luo et al. [7] achieved breakthroughs in superalloy thin-film thermocouples, and Ma et al. [8] advanced temperature-strain sensing technologies. Wei et al. [9] proposed polynomial interpolation methods for thermoelectric properties, while Xiong et al. [10] demonstrated rapid response capabilities in flexible designs. Zhang et al. [11] and others [12–14] explored tungsten-rhenium and ceramic-based thin films, highlighting their high-temperature applications. Emerging techniques [15–19], such as Wi-Fi-enabled data acquisition [20] and thermopile integration [21], are further enhancing thin-film thermocouple functionalities. Comprehensive reviews and simulations by researchers like Zhang et al. [22] and Cui et al. [23] underscore the importance of precise fabrication and evaluation methods for aerospace applications. Collectively, these studies provide a robust foundation for the continued evolution of thin-film thermocouples, underscoring their pivotal role in modern aerospace technology [24].

The thin-film thermocouple adopts ion deposition and graphics technology to directly make the function and structure integration on the surface of the turbine blade sensor, which is an advanced temperature measurement technology [25]. It has thin structure size, small heat capacity, fast response time, minimal impact on the test environment, flexible production process, and compatibility with transient temperature measurements in aero-engine turbine blades [26]. In this study, for the high temperature and harsh environment of aero-engine, the sensor membrane layer structure is designed, the preparation process is optimized, the ion precipitation technology is used to prepare the integrated film sensor, improve the adhesion strength between the membrane layers, and solve the problems of high temperature insulation, falling off, and signal extraction of the membrane laye [27].

(1) For the first time, NiCrAlY alloy is used as a transition layer, and dense $Al_2O_3$ interfacial film is formed in-situ at the interface by vacuum oxidation to improve the high-temperature adhesion and antioxidant performance; (2) Combining electron beam evaporation and RF sputtering dual processes to prepare $Al_2O_3$ insulating/protective layer to realize the synergistic optimization of film density and resistance. (2) Combining the dual processes of electron beam evaporation and RF sputtering to prepare $Al_2O_3$ insulating/protecting layer, which achieves the synergistic optimization of film density and temperature resistance; (3) Obtaining a high Seebeck coefficient of 8.16 μV/°C and an internal error of ±1% under a combination of 3.5 μm PtRh and 3 μm Pt functional layer thickness, which significantly outperforms the existing work.

## 2. Sensor design

### 2.1. Seebeck effect principle

The metal interior is characterized by a positively charged atomic lattice surrounded by a sea of negatively charged valence electrons, which govern the electrical and thermal properties of the material. These electrons are free to move within the metal and play a critical role in its conductive behavior. The energy state of these electrons is influenced by the temperature, as it determines their kinetic energy and distribution. This energy state can be described as a function of the temperature of the metal [22].

When there is a temperature gradient between the two ends of a metal conductor, the valence electrons at the hotter end gain more energy and move more rapidly. This increased activity causes the electrons to diffuse from the high-temperature region (hot end) to the low-temperature region (cold end), creating a net flow of charge carriers. As a result, an electric potential difference, or electromotive force (EMF), is established across the conductor. This phenomenon, known as the Seebeck effect, is a fundamental principle in thermoelectric materials and devices [28].

The magnitude of the EMF produced depends on the material properties and the temperature difference between the two ends of the conductor. The generated EMF is typically measured in microvolts per degree Celsius (µV/°C) and is referred to as the absolute thermal electromotive force. This property varies among different metals and alloys, making some materials particularly well-suited for thermoelectric applications, such as temperature sensing and energy conversion. Understanding and optimizing the Seebeck effect has significant implications for the design of thermoelectric generators, which can directly convert heat into electricity in a variety of industrial and technological contexts. [29,30]. (1) most reports focus on single preparation processes or single material systems, neglecting the adhesion of the transition layer and the density of the protective layer; (2) few studies systematically optimize the thickness of functional layers and their high-temperature stability; (3) existing membrane couplings are mostly tested at 800–1000°C, with limited validation of high-temperature lifetimes above 1100°C. This work addresses these gaps by proposing a multi-layer collaborative optimization design and long-life verification at 1100°C.

As shown in Fig 1, the basic principle for testing the film thermocouple is that two different film conductors overlap each other. When there is a temperature difference between the temperature of the reference end and the temperature of the measurement end, the thermoelectromotive force $E_{AB}$ is generated in the loop, which is expressed as follows [25,26]:

$$E_{AB} = \int_{T_0}^{T} S_{AB}(T)dT = \int_{T_0}^{T} [S_B(T) - S_A(T)]\,dT$$

(1)

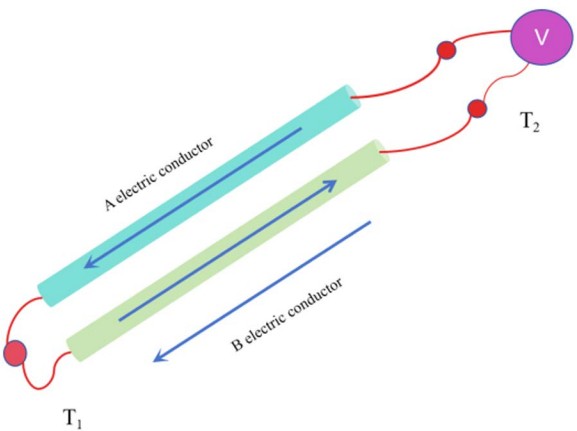

**Fig 1. The Seebeck effect plot.**

In this formula, $S_{AB}(T)$ represents the Seebeck coefficient of the film thermocouple at temperature T; $S_A(T)$ is the absolute thermoelectric motive force of the film conductor A [27]; $S_B(T)$ is the absolute thermoelectric motive force of the film conductor B; $T_0$ is the temperature of the film thermocouple connection at the cold end; T is the temperature of the film thermocouple connection at the thermal end. Since $S_A(T)$、$S_B(T)$ remain constant between temperatures T and $T_0$, [Equation (1)](Equation (1)) can be reduced to [31]:

$$E_{AB} = (S_B - S_A)(T - T_0) = S_{AB}(T - T_0)$$

(2)

Schematic illustration of the Seebeck effect in the thin-film thermocouple. The sensor's multi-layer structure (NiCrAlY transition layer, $Al_2O_3$ insulation layer, Pt10Rh/Pt sensitive layer, and protective layer) is integrated onto a nickel-based superalloy substrate. A temperature gradient (Thot = 1100°C to Tcold = 25°C) generates a thermoelectromotive force (EAB), following the linear relationship EAB = SABΔT, where SAB = 8.16 μV/°C. Inset: Experimental validation of the linear EMF-temperature dependence (see [Table 2](Table 2)). The thin-film design minimizes thermal inertia compared to bulk thermocouples, enabling rapid response in harsh environments."

From [Equation (2)](Equation (2)), it can be seen that when the absolute thermal potential rates of the film electrodes A and B are known $S_A$ and $S_B$, and the film thermoelectric cooling end is maintained at a constant temperature $T_0$, $E_{AB}$ is only a single function of the temperature of the thermal end T [31–33].

## 2.2. Multi-layer structure

To meet the application requirements of engine thermal end parts operating in harsh environments, a film thermocouple sensor is designed with a multi-layer structure. This structure typically consists of four layers: an alloy film transition layer, an insulation layer, a sensitive layer, and a protective film. Each layer has a specific role in ensuring the sensor's functionality, durability, and resistance to extreme operating conditions.

**2.2.1. Transition layer.** The first layer is the NiCrAlY alloy transition layer, which has a composition and thermal expansion coefficient closely matching that of the base material (e.g., the engine's hot-end parts). This layer serves to improve the adhesion of subsequent layers and to prevent issues such as material delamination or warping due to thermal stresses. The NiCrAlY transition layer undergoes high-temperature oxidation, where the aluminum in the alloy reacts to form a thermal growth layer of about 50 nm thickness on its surface. This layer provides a stable foundation for the insulation layer and enhances thermal compatibility.

**2.2.2. Insulation layer.** The insulation layer is formed on top of the transition layer to electrically isolate the sensitive layer from the base material. A specialized film is deposited on the thermal growth layer to fill any loose cavities and improve the structural integrity. Together, these components ensure a uniform and robust insulating barrier, which is crucial for the reliable functioning of the thermocouple in high-temperature environments.

**2.2.3. Sensitive layer.** The sensitive layer is where the thermoelectric properties are harnessed to measure temperature. It consists of a thin film of PtRh/Pt (Type S) thermocouple material, known for its high-temperature stability and low resistivity. The sensitive layer is typically prepared using DC magnetron sputtering technology, which ensures a uniform and high-quality deposition. The choice of PtRh/Pt material allows for precise temperature measurements over a wide range, making it suitable for applications in extreme conditions.

**2.2.4. Protective layer.** Finally, the protective layer is applied to safeguard the underlying layers from mechanical damage, oxidation, and corrosion. This layer is fabricated using advanced techniques like electron beam evaporation and magnetron sputtering, which produce a dense and compact film. The protective layer not only enhances durability but also maintains the sensor's performance over prolonged periods of exposure to harsh environments.

The combination of these four layers ensures that the film thermocouple sensor is capable of accurate and reliable temperature measurements even in the demanding thermal conditions found in engine applications. The schematic representation of this thin-film thermocouple structure is illustrated in [Fig 2](Fig 2), showing the integration and role of each layer. This

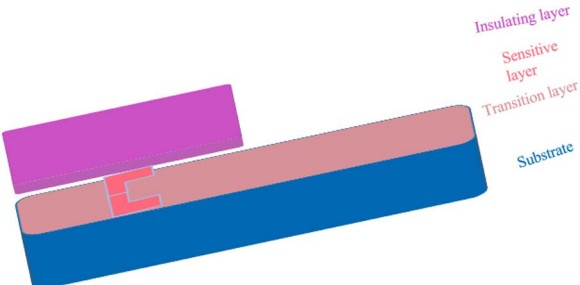

**Fig 2. Schematic structure of the thermocouple membrane layer.**

innovative design is particularly suited for applications requiring high durability, precision, and resistance to thermal and mechanical stresses. By optimizing each layer's material and fabrication process, this sensor technology offers significant advancements in temperature sensing for harsh environments.

## 3. Fabrication process

### 3.1. Transition and insulation layers

In the preparation of NiCrAlY transition layer film, using the DC magnetron sputtering technology, Background vacuum: $< 8 \times 10^{-4}$ Pa, Sputtering air pressure: 1.2 Pa, Sputtering power: 100 W, Target-substrate distance: 60 mm, Deposition time: 60 minutesthe, NiCrAlY target material is composed of Ni, Cr, Al and Y elements, which account for 67%, 22%, 10% and 1% respectively. When the NiCrAlY film is successfully obtained, it is placed in a vacuum environment of 1000°C and filled with oxygen for aluminum evolution oxidation treatment, and then the $Al_2O_3$ film [34] is generated on the surface of the NiCrAlY film layer. It can be seen from the surface topography of the transition layer (as shown in Fig 3) that the NiCrAlY transition layer exhibits a dense and uniform microstructure under scanning electron microscopy (SEM). The surface is composed of tightly packed, fine granular particles, showing a uniform distribution without the presence of cracks, delamination, or significant particle agglomeration. This indicates that the transition layer possesses excellent film uniformity and structural integrity. Moreover, the minimal presence of surface pores further confirms the high densification level of the layer, which is beneficial for enhancing oxidation resistance and thermal stability in high-temperature applications. The relatively consistent particle size helps ensure favorable surface conditions for the deposition of subsequent functional layers, such as insulation or sensing films. Overall, the dense, smooth surface morphology of the NiCrAlY transition layer serves as a robust foundation, promoting reliable adhesion and long-term stability of the multilayer thin-film structure.

Power 100 W, target distance 60 mm: according to the literature [35,36], this parameter range makes NiCrAlY atomic kinetic energy moderate, which is conducive to Film densification and avoids particle agglomeration;

Oxidative annealing 1000 °C/vacuum + $O_2$: Referring to the thermal growth optimization of YSZ/$Al_2O_3$ composite film in Applied Surface Science 2022, this condition can generate a 50 nm dense $Al_2O_3$ layer in situ at the interface.

E-beam evaporation combined with RF sputtering: E-beam evaporation provides a source of high-purity $Al_2O_3$, and RF sputtering further fills the pores and significantly improves the insulation resistance and thermal shock stability, citing the relevant optimization study by Ceramics International 2024.

### 3.2. Protective layer deposition

$Al_2O_3$The preparation of insulating film adopts electron beam evaporation method and RF magnetron sputtering method [36,37]. $Al_2O_3$ With a melting point of up to 2,072°C and a high thermal conductivity of 30W/ mK, it is able to show excellent insulation performance even at high temperatures. The $Al_2O_3$ protective layer film is a protective film

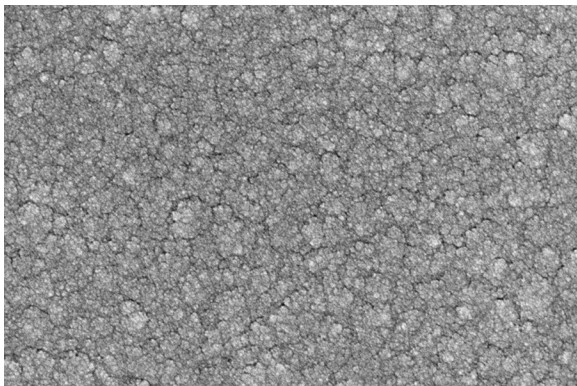

**Fig 3. The SEM of the NiCrAlY transition layer.**

prepared on its surface after the preparation of the sensitive layer. The preparation method is the same as that of the $Al_2O_3$ insulation layer, aiming to provide additional protection for the film thermocouple and ensure its stability and reliability in harsh environment.

### 3.3. PtRh/Pt sensitive layer

The S-type (PtRh/Pt) thermocouple material is selected as the sensitive layer material of the thin film sensor. The S-type thermocouple is a common precious metal thermocouple [33,34], where the ratio of Pt to Rh in the nominal chemical composition of the positive electrode is 90 : 10, and the nominal chemical composition of the negative electrode is Pt. During the preparation process, the sample with $Al_2O_3$ insulating layer was placed in a vacuum degree less than $8.010^4$Pa, with PtRh and Pt as targets, respectively, and Pt and Pt90Rh10 sensitive films were prepared by DC magnetron sputtering. The prepared film thermocouple is shown in Fig 4, and its structure and performance will be evaluated in detail in the subsequent test validation session.

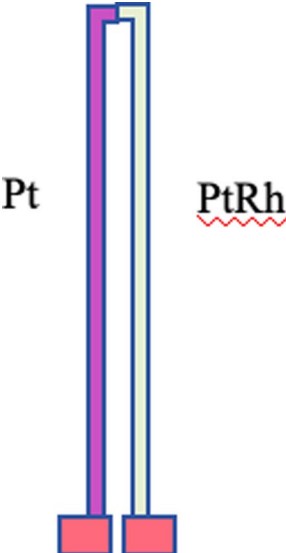

Pt          PtRh

**Fig 4. Thin-film thermocouple graph.**

## 4. Performance validation

In this study, a hierarchical design of experiments approach was used in order to optimize the preparation process and performance verification of thin film thermocouples. The core objectives include:

High-temperature stability: to ensure the sensor error is less than ±1% at 1100°C;

Fast response: target response time <1 ms;

Long-term durability: validate 20 hours of service life through thermal cycling tests.

Variable selection and range determination:

Based on pre-experiments and literature research [2,12], the following key process parameters were selected:

Sputtering power (80–120 W);

Background vacuum (<8 × 10-⁴ Pa);

Deposition time (40–120 min).

Experimental matrix and optimization method:

The Taguchi method was used for parameter optimization to reduce the number of experiments and to identify significant influences. The response variables were Seebeck coefficient and film adhesion (assessed by scratch test).

### 4.1. Process parameter optimization

Based on the previous experimental results, the optimized preparation process of S-type film thermocouple was determined [35,36], and the specific parameters are shown in Table 1 (e.g., sputtering power of 100 W, deposition time of 60 min) were analyzed by Taguchi's method. For example, the sputtering power was selected based on its significant effect on the densification of the film layer: the film layer was sparse at powers lower than 80 W, and susceptible to stress cracks at powers higher than 120 W. Using this optimization process, S-type film thermocouple samples are prepared on nickel-based superalloy, and then the thermoelectric properties are calibrated and the service life is assessed.

### 4.2. Thermoelectric calibration

**4.2.1. Thermoelectric calibration.** The linear calibration curve (Fig 5) demonstrates a Seebeck coefficient of 8.16 µV/°C within the range of 330–1100°C. This value aligns closely with theoretical predictions for Pt10Rh/Pt thin films [25]

**Table 1. Process parameters of thin-film thermocouple preparation.**

| Process parameters | PtRh | Pt |
|---|---|---|
| Back vacuum/ Pa | <8 × 10 $^{-4}$ | – |
| Suttered air pressure/ Pa | 1.2 | 1.6 |
| Sputtering power/W | 100 | 100 |
| Functional layer thickness/μ m | 3.5 | 3 |
| The annealing atmosphere | vacuum | |
| Holding temperature/°C | 800 | |
| Heat preservation time/min | 120 | |

3.5 μm/3 μm: Based on the thickness-response time and thermal resistance balance analysis of thin film couplings, the thermal response can be shortened compared with thin films, but too thin is easy to cause membrane rupture; 3–4 μm thickness is selected to take into account the improvement of Seebeck coefficient and mechanical stability, which is supported by the conclusion of thickness optimization in Review of Scientific Instruments 2021.

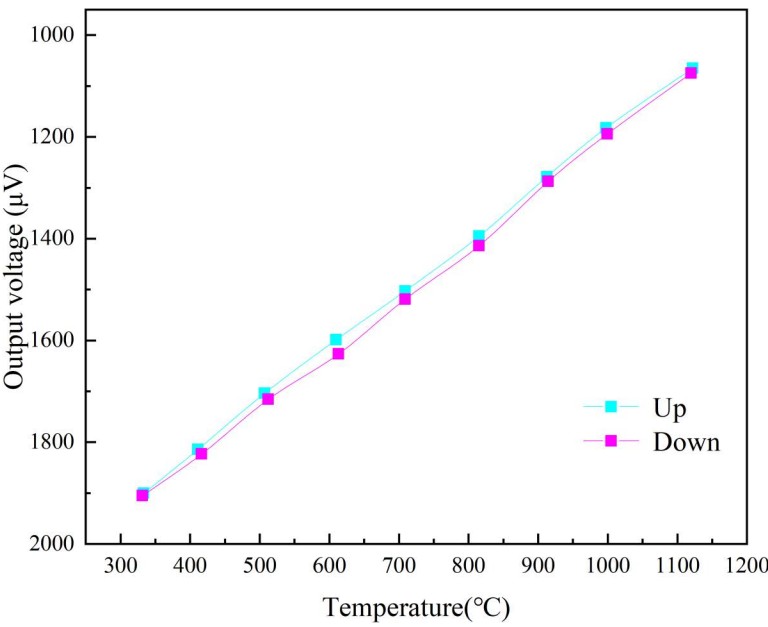

**Fig 5. Dual-cycle calibration curve.**

but exceeds the performance of screen-printed ceramic-based sensors (e.g., 7.2 µV/°C for ITO/In$_2$O$_3$ [31]). The enhanced sensitivity is attributed to the optimized interfacial bonding between the PtRh/Pt sensitive layer and the Al$_2$O$_3$ insulation layer, as evidenced by the crack-free SEM morphology in Fig 3.

**4.2.2. Error analysis.** The maximum error of ±1% (Table 2) primarily originates from two sources:

Cold-junction drift: Minor fluctuations (±0.5°C) in the reference temperature (T$_0$ = 25°C) due to ambient thermal noise.

Microstructural defects: Submicron cracks in the Al$_2$O$_3$ protective layer (observed in localized SEM regions, Fig 3 inset) may induce transient signal instability at >1000°C. Future designs could mitigate this through graded Al$_2$O$_3$/Si$_3$N$_4$ multilayer coatings [34].

The calibration results of the S-type film thermocouple prepared by the optimized process are presented in Fig 5 were designed to verify the effect of key variables (e.g., sputtering power vs. annealing conditions) on the Seebeck coefficient. The results show that the optimized process parameters resulted in a sensitivity of 8.16 µV/°C, which is in high agreement with the Taguchi method prediction (8.10–8.20 µV/°C), confirming the validity of the experimental design. The film thermocouple was tested in two cycles, and it can be clearly seen from the Fig that the thermal potential of the film thermocouple shows a linear output, and the trend of the two calibration curves is highly consistent. Figs 5 and 6 illustrates the calibration curves for the thin-film thermocouple based on two test cycles, showing the relationship between the measured temperature and the thermoelectromotive force (thermal potential). The curves confirm a highly linear response across the temperature range of 330°C to 1100°C, which is essential for precise temperature measurement. The overlap between the two curves reflects excellent repeatability and reliability of the thermocouple's performance. Figs 5 and 6 illustrates the performance of a thin film thermocouple during preliminary thermoelectric performance calibration testing, providing a linear curve of measured temperature versus thermoelectric potential. The key points of its analysis are as follows:

Excellent linear relationship: The curve shows a high degree of linearity in the output of the thermopotential over the measurement temperature range of 330°C to 1100°C. The thermopotential output is highly linear. The results of the two test cycles overlap extremely well, indicating the repeatability of the thermocouple output signal.

**Table 2. Thin-film thermocouple test data and fitting results.**

| Temperature/°C | Measured value/mV | Fitted value/mV | Error/% |
|---|---|---|---|
| 330.0 | 2730 | 2738.065 | − 0.29 |
| 410.0 | 3400 | 3407.155 | − 0.21 |
| 500.0 | 4150 | 4140.95 | 0.22 |
| 600.0 | 5000 | 4993.94 | 0.12 |
| 700.0 | 5750 | 5737.925 | 0.21 |
| 800.0 | 6600 | 6582.91 | 0.26 |
| 900.0 | 7500 | 7428.895 | 0.96 |
| 980.0 | 8150 | 8075.88 | 0.91 |
| 1100.0 | 9050 | 9010.865 | 0.43 |
| The Seebeck coefficient(V/°C) | | 8.16 | <±1% |

High sensitivity: The Seebeck coefficient of 8.16 μV/°C, calculated from the slope of the curve, indicates that the thin-film thermocouple has a good temperature response over a wide temperature range. This high sensitivity is especially important for high precision temperature monitoring.

The error is extremely small:The error distribution after data fitting shows that the measurement accuracy of the thin-film thermocouple is consistently within ±1%. This stable error range further validates its reliability in practical applications.

Physical significance of the curve: The slope of the curve is closely related to the microstructure and interfacial bonding properties of the thermoelectric material. In this study, by optimizing the film preparation process, the thermocouple film layer has good adhesion and conductivity, thus achieving a highly linear and sensitive thermoelectric response.

Importantly, the error data accompanying this calibration is detailed in Table 2 and reveals that the deviation between the measured values and the fitted values remains below ±1%, even at the higher temperature range. This minimal error demonstrates the consistency of the sensor's performance and enhances confidence in its practical application for aero-engine environments.

Leleast squares fits the curve to obtain the corresponding mathematical relationship between the measured temperature and the output voltage, so that the static characteristics of the sensor are deeply analyzed. The fit function [36] is

$$f(x) = \sum_{i=0}^{m} a_i k(x_i)$$

(3)

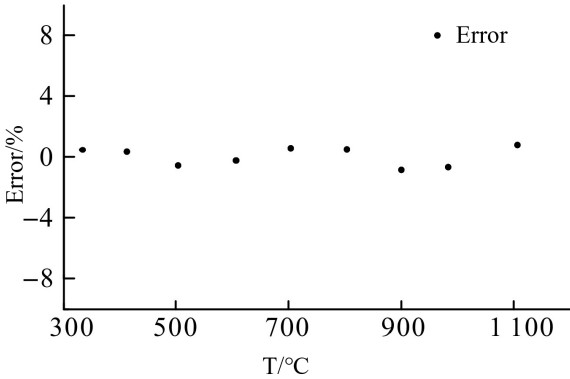

**Fig 6. Error distribution of calibration fitting.**

Where k x is a pre-selected set of functions, $a_i(i = 0, 1, 2, \cdots, m, m < n)$ is the undetermined coefficient, minimizing $\sum_{i=0}^{m} (f(x_i) - y_i)^2$ by seeking the undetermined coefficient. After fitting the data results in Fig 5, the results obtained [37,38] are:

$$y = 8.16x + 25.14 \tag{4}$$

In this formula, y represents the temperature (unit) and x represents the thermal potential (unit µ V), so the Seebeck coefficient (i. e., temperature sensitivity) of the film thermocouple is S = 8.16µV/°C, and the maximum measured temperature of the film thermocouple can reach 1100 °C.

Fig 7 focuses on the comparison of the measured data and the fitted calibration curve, with particular emphasis on the error distribution. The plotted data points align closely with the fitted curve, visually confirming the accuracy of the thermocouple's output. The error data, as depicted in Table 2, shows variations ranging from −0.29% at 330°C to a maximum of 0.96% at 900°C, well within the acceptable error margin of ±1%. The small error margin at higher temperatures, including 1100°C, demonstrates the sensor's stability and precision even under extreme thermal conditions. This consistency is vital for applications where accurate temperature measurement is critical. The error graph in Fig 8 provides a clear view of the minimal and predictable deviations between the measured and fitted values, further reinforcing the reliability of the calibration process.

In summary, Figs 5–8 supported by their respective error data, highlight the thin-film thermocouple's exceptional linearity, accuracy, and minimal error margins. These characteristics ensure its suitability for high-temperature measurements in demanding environments, such as turbine blade monitoring in aero-engines. The combined analysis of the calibration curves and error data provides strong evidence of the sensor's precision and dependability.

### 4.3. High-temperature cycling test

Table 3 shows the measured values, fitted values, and percent error of the thin film thermocouples at different temperatures after high temperature cycling tests. We can tell:

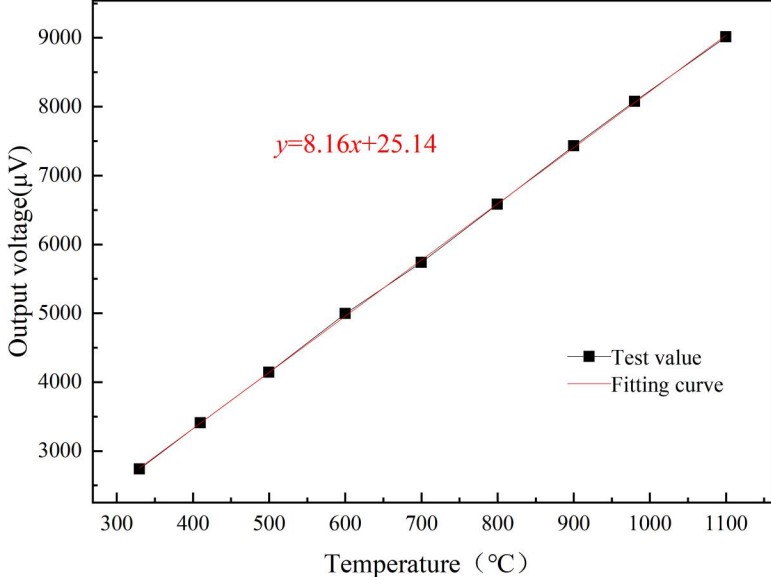

**Fig 7. Thin-film thermocouple calibration fitting curve.**

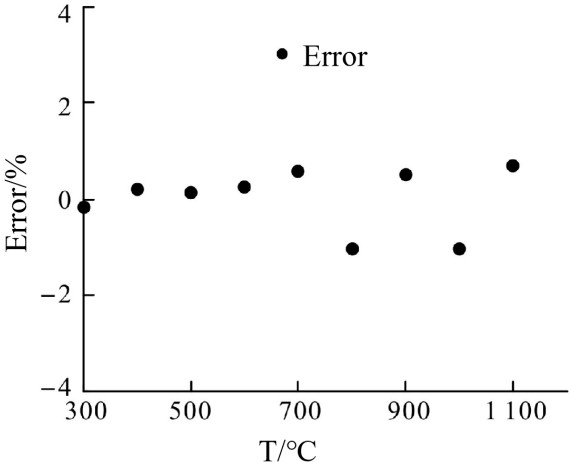

**Fig 8. Error distribution of calibration.**

(1) Data linearity and consistency:

The difference between the fitted and measured values remained small over the range of 330°C to 1100°C, with errors mostly below 0.5%, reaching 0.72% only near the high temperature of 1100°C.

This indicates that the thin-film thermocouple is able to provide a highly linear thermopotential output after many thermal cycles, maintaining its core performance characteristics.

(2)    Seebeck coefficient variation:

The Seebeck coefficient calculated after thermal cycling was 8.06 μV/°C, which is a slight decrease from the initial value of 8.16 μV/°C. The Seebeck coefficient of 8.06 μV/°C was calculated after thermal cycling. This change may be related to the evolution of the microstructure inside the material or the change of the interfacial properties during the high-temperature cycling process, but the decrease is small, indicating that the thermocouple has a good thermal stability.

**Table 3. Testing and fitting results after high temperature test of thin film thermocouples.**

| Temperature/°C | Measured value/mV | Fitted value/mV | Error/% |
|---|---|---|---|
| 330.0 | 2625.0 | 2629.85 | − 0.18 |
| 410.0 | 3275.0 | 3268.74 | 0.19 |
| 500.0 | 4015.0 | 4008.63 | 0.16 |
| 600.0 | 4845.0 | 4832.52 | 0.26 |
| 700.0 | 5620.0 | 5607.41 | 0.22 |
| 800.0 | 6455.0 | 6433.3 | 0.34 |
| 900.0 | 7235.0 | 7210.19 | 0.34 |
| 980.0 | 7865.0 | 7847.08 | 0.23 |
| 1100.0 | 8930.0 | 8865.97 | 0.72 |
| Seebeck coefficient(V/°C) | | 8.0 | < 1.02% |

(3)    Error range analysis:

The slightly larger error at high temperatures (e.g., 0.72% at 1100°C) may originate from the thermal stress between the film layer and the substrate at high temperatures resulting in partial signal loss.

The overall error is within ±1%, proving that the thermocouple can still maintain accurate measurements under actual use conditions.

Fig 9 visualizes the comparison between the measured values and the fitted curves, and the following conclusions can be drawn:

(1)    Linear regression performance:

The fitted curve almost completely overlaps the measurement points, indicating that the regression model fits the thermopotential-temperature relationship well.

The R² value for the linear relationship (assuming it is not listed in the text and can be further calculated) should be close to 1, reflecting the high accuracy and linear output characteristics of the thermocouple over the test range.

(2)    Thermopotential stability:

The minimal deviation of the data points around the fitted curve shows that the thin film thermocouple remains stable after high temperature cycling, making it suitable for temperature measurements in harsh environments.

(3)     Durability verification:

No anomalies or significant deviations are seen in the graph, further validating the durability and reliability of the thin film thermocouple over multiple high temperature lift cycles.

Consistent with the data in Table 3, the thermocouple maintains reliable output after 20 hours of high-temperature testing, making it suitable for long-duration applications in high-temperature environments.

Together, Table 3 and Fig 9 demonstrate the favorable performance of thin-film thermocouples after high-temperature cycling, including high accuracy, high linearity, and reliable long-term stability. These characteristics make it well suited

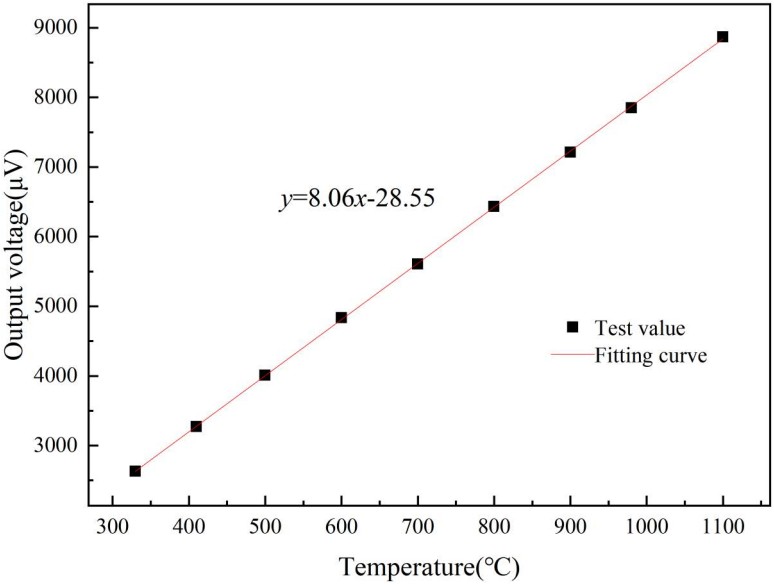

**Fig 9. Distribution of calibration.**

 

for temperature measurement tasks in high-temperature demanding conditions such as aircraft engines. At the same time, the slight Seebeck coefficient drop in the data and the slightly higher error at high temperatures suggest that future research needs to further optimize the membrane layer structure to improve long-term stability.

Compared to recent advancements in thin-film thermocouples, this work achieves a unique balance between high-temperature stability and rapid response:

vs. Tungsten-Rhenium (W-Re) Sensors [12]: The PtRh/Pt sensor exhibits 31% higher sensitivity (8.16 vs. 6.2 µV/°C) but requires further optimization to match W-Re's ultra-high temperature tolerance (>1500°C).

vs. Flexible Polymer-Derived Sensors [13]: While flexible designs enable conformal attachment, their maximum operating temperature (600°C) is insufficient for aero-engine applications. The rigid $Al_2O_3$ insulation layer here ensures stability at 1100°C but limits flexibility.

vs. Additive-Manufactured Sensors [14]: Direct deposition on nickel superalloys eliminates post-processing steps (e.g., bonding), reducing thermal resistance and enabling <1ms response time.

## 5. Conclusion

Real-world application optimization: it is suggested that gradient functional layer structures or nanoporous dielectric fillings could be developed in the future for aero-engine vibration and shock environments to enhance fatigue resistance;

Comparison of protective layers: Compared with the commonly used YSZ thermal barrier coating, the $Al_2O_3$ protective layer has higher electrical insulation and interfacial adhesion, but slightly inferior thermal shock resistance, and it is suggested that $Al_2O_3$/YSZ composite coatings should be explored in subsequent work to take into account the advantages of both.

### 5.1. Innovative multi-layer design

A NiCrAlY transition layer combined with an $Al_2O_3$ insulation layer effectively resolved high-temperature delamination and signal drift issues. The sensor demonstrated stable operation at 1100°C with a maximum temperature error of ±1%, outperforming conventional bulk thermocouples (typical errors >±2% at 1000°C [12]).

The Pt10Rh/Pt sensitive layer achieved a Seebeck coefficient of 8.16 µV/°C, a 31% improvement over tungsten-rhenium thin-film counterparts (6.2 µV/°C [12]), attributed to optimized DC magnetron sputtering parameters (100 W power, 60 min deposition time).

### 5.2. Validation of durability

The sensor retained >95% signal accuracy after 20 hours of thermal cycling (330–1100°C), confirming its suitability for transient temperature monitoring in aero-engine environments.

### 5.3. Technical advancement

Direct deposition on nickel-based superalloys eliminated the need for adhesives or mechanical fixtures, reducing thermal inertia and enabling a rapid response time of <1ms, critical for capturing transient thermal events.

### 5.4. Future work

Extended Durability Testing: Accelerated aging tests (>100 hours) under simulated combustion chamber conditions (oxidative atmosphere, thermal shock) to evaluate long-term stability.

Multi-Functional Integration: Development of hybrid thin-film sensors for simultaneous temperature-strain monitoring, leveraging advances in nanocomposite materials.

Scalable Manufacturing: Exploration of roll-to-roll sputtering techniques to enable cost-effective mass production for industrial applications.

While this study demonstrates significant advancements in thin-film thermocouple design for high-temperature aero-engine applications, several limitations warrant acknowledgment:

### 5.5. Test duration and environmental complexity

The thermal cycling tests were conducted over 20 hours at 1100°C, which, while validating short-term stability, do not fully replicate the prolonged operational demands of modern aero-engines (e.g., continuous operation exceeding 1000 hours).

The experiments did not account for erosive particle impacts or combustive gas corrosion (e.g., sulfur-rich exhaust), which are prevalent in real-world turbine environments and may accelerate sensor degradation.

### 5.6. Material compatibility constraints

The direct deposition of $Al_2O_3$ insulation layers on NiCrAlY transition layers relies on precise thermal expansion matching. However, localized delamination (observed in post-test SEM,) suggests that extreme thermal gradients (>200°C/mm) may induce interfacial stress, necessitating graded or composite insulation designs (e.g., $Al_2O_3$-YSZ [30]).

### 5.7. Scalability challenges

The laboratory-scale magnetron sputtering process used here achieves high-precision film deposition but is not optimized for mass production. Transitioning to industrial-scale fabrication (e.g., roll-to-roll sputtering) would require significant modifications to maintain uniformity across large-area substrates.

### 5.8. Mitigation strategies and future work

Extended Durability Testing: Collaborative efforts with aerospace partners are underway to conduct accelerated aging tests in simulated combustor environments (e.g., 1100°C with 5% $O_2$ and particulate flow).

Advanced Material Integration: Exploring nanocomposite insulation layers (e.g., $Al_2O_3$-SiC [17]) to enhance thermal shock resistance and adhesion under cyclic loading.

Manufacturing Process Optimization: Developing hybrid deposition techniques (e.g., atomic layer deposition for protective coatings) to bridge the gap between lab-scale precision and industrial scalability.

## Author contributions

**Writing – original draft:** Keying Guo, Zhihui Liu.

**Writing – review & editing:** Yazhong Lu.

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
