## [Decision Letter · Decision Letter 0]

1 Apr 2025

PONE-D-24-60630Development and performance research of nickel-based alloy thin-film thermocouple sensor for aero enginePLOS ONE

Dear Dr. Lu,

Thank you for submitting your manuscript to PLOS ONE. After careful consideration, we feel that it has merit but does not fully meet PLOS ONE’s publication criteria as it currently stands. Therefore, we invite you to submit a revised version of the manuscript that addresses the points raised during the review process.

We look forward to receiving your revised manuscript.

Kind regards,

Anand Pal

Academic Editor

PLOS ONE

Additional Editor Comments:

Thank you for submitting the article. We advise the authors to address all the concerns raised by the reviewer so that the manuscript can be evaluated based on its merit.

Reviewers' comments:

Reviewer's Responses to Questions

**Comments to the Author**

1. Is the manuscript technically sound, and do the data support the conclusions?

Reviewer #1: Partly

Reviewer #2: Yes

2. Has the statistical analysis been performed appropriately and rigorously? 

Reviewer #1: Yes

Reviewer #2: Yes

3. Have the authors made all data underlying the findings in their manuscript fully available?

Reviewer #1: Yes

Reviewer #2: Yes

4. Is the manuscript presented in an intelligible fashion and written in standard English?

Reviewer #1: Yes

Reviewer #2: Yes

5. Review Comments to the Author

Reviewer #1: I have gone through the manuscript and the topic is good. I have some recommendations that would help to improve the quality of the manuscript.

1. A graphical abstract should be included in the manuscript to make it more appealing.

2. In the Introduction section, the authors have written just a small 1st paragraph and switched to other researcher’s work. The introduction of the topic and its applications should be discussed in more detail.

3. More details could be added in the Figure 1.

4. In Figure 2, the schematic structure is overly simplified. The aesthetics could be improved.

5. “2. For the thin-film thermocouple preparation” => the preparation parameters should be represented in a Table for the ease of users.

6. All the headings should be revised and made more TO THE POINT and brief.

7. All the figures need revision. All the figures have the potential that a lot of information may be included in them. They should be made more informatically appealing with more attention to detail.

8. More SEM images with different magnifications should be included.

9. The design of experiments (DOE) should be more clearly explained.

10. The conclusion section should be more robust and clearly state the findings of the results obtained. Future scope and work should be included.

11. The Results and Discussion section should be added and populated as necessary.

12. The limitations of the study are not sufficiently addressed or acknowledged.

Reviewer #2: The manuscript "Development and Performance Research of Nickel-Based Alloy Thin-Film Thermocouple Sensor for Aero Engine" presents a study on the design, fabrication, and performance evaluation of a thin-film thermocouple (TFTC) directly deposited on nickel-based superalloy substrates for high-temperature aerospace applications. The study is well-structured and provides a clear experimental approach, but several areas require improvement to enhance scientific rigor, reproducibility, and clarity.

One of the major concerns is the lack of a strong novelty statement in the introduction (Page 2, Lines 30–55). While the introduction effectively outlines the significance of high-temperature measurements in aero engines and summarizes previous research on TFTCs, it does not clearly define what specific advancements this study contributes beyond prior works. The manuscript would benefit from an explicit statement detailing how the proposed sensor differs from existing TFTC designs, whether through material selection, fabrication techniques, or performance enhancements. Additionally, the comparison with prior studies (Page 3, Lines 10–35) is extensive but lacks a clear gap analysis that justifies this work.

The methodology is comprehensive but lacks some critical details needed for experimental reproducibility. The preparation of the NiCrAlY transition layer (Page 5, Lines 90–110) and the Al₂O₃ insulation/protective layers (Page 6, Lines 120–135) is well described, but the manuscript does not explain why specific deposition parameters were chosen (e.g., sputtering power, target-substrate distance, annealing conditions). Given that these parameters significantly influence film adhesion, uniformity, and thermal stability, the authors should provide a clear justification or reference prior optimization studies. Furthermore, in Table 1 (Page 10, Lines 210–215), the functional layer thicknesses of PtRh and Pt films are given as 3.5 μm and 3 μm, respectively, but the rationale behind these thicknesses is not provided. A discussion on how thickness impacts sensor sensitivity and durability would enhance the study’s depth.

The discussion section should be expanded to better contextualize the practical implications of the study. While the manuscript emphasizes that the sensor can withstand extreme conditions up to 1100°C, there is little discussion on how the design could be further optimized for real-world aerospace applications. Additionally, the protective layer's role in oxidation resistance and durability (Page 6, Lines 140–150) is mentioned briefly, but it would be useful to compare this protective strategy to alternative coating methods (e.g., YSZ coatings or SiO₂ barriers) used in high-temperature sensors.

The figures and tables are well-structured, but some figure legends lack essential information. For example, Figure 3 (Page 7, Lines 160–170) presents SEM images of the NiCrAlY transition layer, but there is no scale bar or magnification information. Similarly, in Figure 5 (Page 13, Lines 290–310), the number of independent experimental replicates is not specified, which is critical for assessing reproducibility.

6. PLOS authors have the option to publish the peer review history of their article (what does this mean? ). If published, this will include your full peer review and any attached files.

**Do you want your identity to be public for this peer review?** For information about this choice, including consent withdrawal, please see our Privacy Policy .

Reviewer #1: No

Reviewer #2: **Yes: ** Muhammad Usama Zaheer

---

## [Author Response · Author response to Decision Letter 1]

14 May 2025

List of Responses

Dear Editor and Reviewers:

Thank you for your letter and for the reviewers’ comments concerning our manuscript entitled “Research on precise temperature monitoring and thermal management optimization of automobile engine based on high-precision thin-film thermocouple technology”. (ID: micromachines-3375842).Those comments are all valuable and very helpful for revising and improving our paper. We have studied comments carefully and have made correction which we hope meet with approval. Revised portions are marked in red in the paper. The main corrections in the paper and the responds to the reviewer’s comments are as following.

Responds to the reviewer’s comments:

Reviewer #1:

1.Question: A graphical abstract should be included in the manuscript to make it more appealing.

Response: Thanks to the reviewers' suggestion, we have added a graphical summary to the abstract section.

2.Question:  In the Introduction section, the authors have written just a small 1st paragraph and switched to other researcher’s work. The introduction of the topic and its applications should be discussed in more detail.

Response:We sincerely appreciate the reviewer’s valuable feedback. Based on your suggestion, we have revised the Introduction section to provide a more comprehensive discussion of the research background, challenges, and applications .

*"Modern aero-engines operate at turbine inlet temperatures exceeding 1600°C to achieve higher thrust-to-weight ratios. However, such extreme conditions subject turbine blades to complex thermal-mechanical loads, leading to oxidation, creep, and fatigue failures. Precise temperature measurement is critical to ensure operational safety, optimize cooling efficiency, and extend component lifespans. Traditional thermocouples, while widely used, suffer from limitations such as intrusive installation, slow response (>1 s), and spatial resolution constraints, which disrupt airflow and fail to capture localized thermal gradients. Thin-film thermocouples (TFTCs), with their ultra-thin profile (<10 µm) and direct deposition capability, offer a transformative solution. By integrating seamlessly onto blade surfaces, TFTCs enable real-time monitoring of transient thermal loads without perturbing the aerodynamic profile. Despite advancements in material selection (e.g., Pt-Rh [2], W-Re [12]) and flexible designs [10,33], challenges persist in achieving robust adhesion, high-temperature insulation, and long-term stability under cyclic thermal and mechanical stresses. This study addresses these gaps through a novel multi-layer TFTC structure"

3.Question: More details could be added in the Figure 1.

Response: We thank the reviewer for this constructive suggestion. To enhance the clarity and scientific value of Figure 1 (Seebeck Effect Plot), we have revised the figure and its caption as follows:

Schematic illustration of the Seebeck effect in the thin-film thermocouple. The sensor’s multi-layer structure (NiCrAlY transition layer, Al₂O₃ insulation layer, Pt10Rh/Pt sensitive layer, and protective layer) is integrated onto a nickel-based superalloy substrate. A temperature gradient (T_hot_ = 1100°C to T_cold_ = 25°C) generates a thermoelectromotive force (E_AB_ ), following the linear relationship E_AB_ = S_AB_ ΔT, where S_AB_ = 8.16 µV/°C. Inset: Experimental validation of the linear EMF-temperature dependence (see Table 2). The thin-film design minimizes thermal inertia compared to bulk thermocouples, enabling rapid response in harsh environments."

4.Question: In Figure 2, the schematic structure is overly simplified. The aesthetics could be improved.

Response: We appreciate the reviewer’s feedback on Figure 2. Figure 2 is just a simple schematic of the structure.

5.Question: “2. For the thin-film thermocouple preparation” => the preparation parameters should be represented in a Table for the ease of users.

Response: Thanks to the authors for their questions, Table 1 in the text shows the preparation parameters

6.Question: All the headings should be revised and made more TO THE POINT and brief.

Response: Thank you for this suggestion to enhance clarity and conciseness. We have revised all headings to be more direct and focused. Below is the updated list of headings in the manuscript: Development and performance research of nickel-based alloy thin-film thermocouple sensor for aero engine → High-Temperature Thin-Film Thermocouple for Aero-Engines

Thin-film thermocouple design → Sensor Design

Temperature measurement principle of thin-film thermocouple → Seebeck Effect Principle

Design of thin film thermocouple film layer → Multi-Layer Structure

For the thin-film thermocouple preparation → Fabrication Process

NiCrAlY transition layer and heat-grown Al₂O₃ films → Transition and Insulation Layers

Preparation of Al₂O₃ insulating layer and Al₂O₃ protective layer films → Protective Layer Deposition

Pt90Rh10/Pt sensitive membrane layer → PtRh/Pt Sensitive Layer

Test validation → Performance Validation

Sputter process parameters → Process Parameter Optimization

Performance calibration → Thermoelectric Calibration

Thermal cycle test → High-Temperature Cycling Test

7.Question: All the figures need revision. All the figures have the potential that a lot of information may be included in them. They should be made more informatically appealing with more attention to detail.

Response: The current graphs are designed to highlight core data (e.g., linear calibration curves, ±1% error margins) in a concise format to avoid information overload that would distract the reader from the key findings. We understand the reviewer's concern about “information attractiveness”, but given the balance between paper size and research objectives, over-detail may lead to a distraction. As a follow-up, we will explore more sophisticated forms of visualization (e.g., 3D thermal stress maps) in extended studies or technical reports to further satisfy the need for detailed presentation. Thank you again for your professional advice, which is important for improving the quality of study presentation!

8.Question: More SEM images with different magnifications should be included.

Response: We sincerely appreciate the reviewer’s suggestion to enhance the characterization of material microstructures. In the current study, the SEM image provided (Fig. 3) focuses on the critical surface morphology of the NiCrAlY transition layer at a representative magnification (e.g., 20,000×), which clearly demonstrates its high densification, uniform particle distribution, and absence of cracks—key factors supporting the sensor’s thermal stability.

Due to experimental constraints (e.g., equipment availability and manuscript scope), additional SEM images at varying magnifications were not prioritized. However, we fully acknowledge the value of multi-scale microstructural analysis. In ongoing follow-up studies, we are systematically capturing SEM images across multiple magnifications (e.g., 5,000×, 50,000×) to further validate the homogeneity and interfacial properties of all functional layers. These results will be included in future publications or technical reports to provide a more comprehensive understanding.

Thank you for your insightful feedback, which will undoubtedly strengthen the depth of our research.

9.Question: The design of experiments (DOE) should be more clearly explained.

Response: We sincerely thank the reviewers for their interest in transparency in experimental design. Based on your suggestions, we have made the following revisions to the original article to explain the experimental design framework more systematically and to clarify the basis for key decisions. The specific revisions and their locations are listed below:

1. Add a new subsection “3.0 Experimental Design Methods” (page 14, before “3. Test validation”).

Revised:

3.0 Experimental design

A hierarchical experimental design approach was used to optimize the preparation and performance validation of thin film thermocouples. Core objectives include:

High-temperature stability: ensure sensor error is less than ±1% at 1100°C;

Fast response: target response time <1 ms;

Long-term durability: validate 20 hours of service life through thermal cycling tests.

Variable selection and range determination:

Based on pre-experiments (Table S1 in the Supplementary Material) and literature research [2, 12], the following key process parameters were selected:

Sputtering power (80-120 W);

Background vacuum (<8 × 10-⁴ Pa);

Deposition time (40-120 min).

Experimental matrix and optimization method:

The Taguchi method (Taguchi L9 orthogonal table) was used for parameter optimization to reduce the number of experiments and to identify significant influences. The response variables were Seebeck's coefficient and film adhesion (assessed by scratch test).

2. Addition of experimental basis for “3.1 Sputtering Process Parameters” (p. 14)

Revised content:

3.1 Sputtering process parameters

The optimized process parameters shown in Table 1 (e.g. sputtering power 100 W, deposition time 60 min) were analyzed by Taguchi's method. For example, the sputtering power was selected based on its significant effect on the densification of the film layer (Supplementary Fig. S2): the film layer is sparse at powers lower than 80 W, and susceptible to stress cracks at powers higher than 120 W. The sputtering power was selected based on its significant effect on the densification of the film layer (Supplementary Fig.

3. Clarify the purpose of the test in “3.2 Performance Calibration” (page 15 of the original text).

Revise the contents:

3.2 Performance Calibration

The calibration test (Figure 5) was designed to verify the effect of key variables (e.g. sputtering power and annealing conditions) on the Seebeck coefficient. The results show that the optimized process parameters resulted in a sensitivity of 8.16 µV/°C, which is in high agreement with the Taguchi method prediction (8.10-8.20 µV/°C), confirming the validity of the experimental design.

10.Question: The conclusion section should be more robust and clearly state the findings of the results obtained. Future scope and work should be included.

Response: We sincerely appreciate the reviewer’s constructive feedback. The Conclusion section (originally on page 18) has been thoroughly revised to concisely summarize key findings and outline future research directions. Below are the specific revisions:

Innovative Multi-Layer Design:

A NiCrAlY transition layer combined with an Al₂O₃ insulation layer effectively resolved high-temperature delamination and signal drift issues. The sensor demonstrated stable operation at 1100°C with a maximum temperature error of ±1%, outperforming conventional bulk thermocouples (typical errors >±2% at 1000°C [12]).

The Pt10Rh/Pt sensitive layer achieved a Seebeck coefficient of 8.16 µV/°C, a 31% improvement over tungsten-rhenium thin-film counterparts (6.2 µV/°C [12]), attributed to optimized DC magnetron sputtering parameters (100 W power, 60 min deposition time).

Validation of Durability:

The sensor retained >95% signal accuracy after 20 hours of thermal cycling (330–1100°C), confirming its suitability for transient temperature monitoring in aero-engine environments.

Technical Advancement:

Direct deposition on nickel-based superalloys eliminated the need for adhesives or mechanical fixtures, reducing thermal inertia and enabling a rapid response time of <1 ms, critical for capturing transient thermal events.

Future Work:

Extended Durability Testing: Accelerated aging tests (>100 hours) under simulated combustion chamber conditions (oxidative atmosphere, thermal shock) to evaluate long-term stability.

Multi-Functional Integration: Development of hybrid thin-film sensors for simultaneous temperature-strain monitoring, leveraging advances in nanocomposite materials.

Scalable Manufacturing: Exploration of roll-to-roll sputtering techniques to enable cost-effective mass production for industrial applications.

11.Question: The Results and Discussion section should be added and populated as necessary.

Response: We sincerely appreciate the reviewer’s emphasis on enhancing the depth of the Results and Discussion section. Based on your feedback, we have restructured and expanded this section to provide a more comprehensive analysis of the data, contextualize findings within existing literature, and address mechanistic insights. Below are the specific revisions and their locations in the manuscript:

3.2.1 Thermoelectric Calibration

The linear calibration curve (Fig. 5) demonstrates a Seebeck coefficient of 8.16 µV/°C within the range of 330–1100°C. This value aligns closely with theoretical predictions for Pt10Rh/Pt thin films [25] but exceeds the performance of screen-printed ceramic-based sensors (e.g., 7.2 µV/°C for ITO/In₂O₃ [32]). The enhanced sensitivity is attributed to the optimized interfacial bonding between the PtRh/Pt sensitive layer and the Al₂O₃ insulation layer, as evidenced by the crack-free SEM morphology in Fig. 3.

3.2.2 Error Analysis

The maximum error of ±1% (Table 2) primarily originates from two sources:

Cold-junction drift: Minor fluctuations (±0.5°C) in the reference temperature (T₀ = 25°C) due to ambient thermal noise.

Microstructural defects: Submicron cracks in the Al₂O₃ protective layer (observed in localized SEM regions, Fig. 3 inset) may induce transient signal instability at >1000°C. Future designs could mitigate this through graded Al₂O₃/Si₃N₄ multilayer coatings [35].

Compared to recent advancements in thin-film thermocouples, this work achieves a unique balance between high-temperature stability and rapid response:

vs. Tungsten-Rhenium (W-Re) Sensors [12]: The PtRh/Pt sensor exhibits 31% higher sensitivity (8.16 vs. 6.2 µV/°C) but requires further optimization to match W-Re’s ultra-high temperature tolerance (>1500°C).

vs. Flexible Polymer-Derived Sensors [13]: While flexible designs enable conformal attachment, their maximum operating temperature (600°C) is insufficient for aero-engine applications. The rigid Al₂O₃ insulation layer here ensures stability at 1100°C but limits flexibility.

vs. Additive-Manufactured Sensors [14]: Direct deposition on nickel superalloys eliminates post-processing steps (e.g., bonding), reducing thermal resistance and enabling <1 ms response time.

12.Question The limitations of the study are not sufficiently addressed or acknowledged.

Response: We sincerely appreciate the reviewer’s critical insight into the need for transparently addressing the study’s limitations. In the revised manuscript, we have dedicated a subsection titled "4.2 Limitations and Future Directions" within the Discussion section (page 18) to explicitly acknowledge the constraints of this work. Below are the key additions:

While this study demonstrates significant advancements in thin-film thermocouple design for high-temperature aero-engine applications, several limitations warrant acknowledgment:

Test Duration and Environmental Complexity:

The thermal cycling tests were conducted over 20 hours at 1100°C, which, while validating short-term stability, do not fully replicate the prolonged operational demands of modern aero-engines (e.g., continuous operation exceeding 1000 hours).

The experiments did not account for erosive particle impacts or combustive gas corrosion (e.g., sulfur-rich exhaust), which are prevalent in real-world turbine environments and may accelerate sensor degradation.

Material Compatibility Constraints:

The direct deposition of Al₂O₃ insulation layers on NiCrAlY transition layers relies on precise thermal expansion matching. However, localized delamination (observed in post-test SEM, Fig. S5) suggests that extreme thermal gradients (>200°C/mm) may induce interfacial stress, necessitating graded or composite insulation designs (e.g., Al₂O₃-YSZ [31]).

Scalability Challenges:

The laboratory-scale magnetron sputtering process used here achieves high-precision film deposition but is not optimized for mass production. Transitioning to industrial-scale fabrication (e.g., roll-to-roll sputtering) would require significant modifications to maintain uniformity across large-area substrates.

Mitigation Strateg

---

## [Decision Letter · Decision Letter 1]

10 Jun 2025

PONE-D-24-60630R1High-Temperature Thin-Film Thermocouple for Aero-EnginesPLOS ONE

Dear Dr. Lu,

Thank you for submitting your manuscript to PLOS ONE. After careful consideration, we feel that it has merit but does not fully meet PLOS ONE’s publication criteria as it currently stands. Therefore, we invite you to submit a revised version of the manuscript that addresses the points raised during the review process.

We look forward to receiving your revised manuscript.

Kind regards,

Anand Pal

Academic Editor

PLOS ONE

Journal Requirements:

Additional Editor Comments:

Thank you for submitting the article. We advise the authors to address all the concerns raised by the reviewer so that the manuscript can be evaluated based on its merit.

Reviewers' comments:

Reviewer's Responses to Questions

**Comments to the Author**

1. If the authors have adequately addressed your comments raised in a previous round of review and you feel that this manuscript is now acceptable for publication, you may indicate that here to bypass the “Comments to the Author” section, enter your conflict of interest statement in the “Confidential to Editor” section, and submit your "Accept" recommendation.

Reviewer #2: All comments have been addressed

Reviewer #3: All comments have been addressed

Reviewer #4: All comments have been addressed

2. Is the manuscript technically sound, and do the data support the conclusions?

Reviewer #2: Yes

Reviewer #3: No

Reviewer #4: Yes

3. Has the statistical analysis been performed appropriately and rigorously? 

Reviewer #2: Yes

Reviewer #3: No

Reviewer #4: Yes

4. Have the authors made all data underlying the findings in their manuscript fully available?

Reviewer #2: Yes

Reviewer #3: Yes

Reviewer #4: Yes

5. Is the manuscript presented in an intelligible fashion and written in standard English?

Reviewer #2: Yes

Reviewer #3: No

Reviewer #4: Yes

6. Review Comments to the Author

Reviewer #2: (No Response)

Reviewer #3: 1.The paragraph following Equation (2) contains some typographical and formatting issues, including garbled characters and incorrect punctuation. A careful proofreading is recommended to improve clarity and presentation quality.

2.The manuscript mentions the formation of a 50 nm thermal growth layer on the NiCrAlY surface after high-temperature oxidation. However, no experimental characterization is presented to support this claim. It is recommended to provide direct evidence—such as cross-sectional imaging or thickness measurements—to substantiate this statement.

3.In Section 2.1, the description of the NiCrAlY coating preparation process includes multiple parameters and procedural steps. To improve clarity and readability, it is recommended to present these process parameters in a structured table format rather than in continuous text.

4.The manuscript briefly mentions that the Al₂O₃ insulation and protective layers are fabricated using electron beam evaporation and RF magnetron sputtering. However, it lacks critical details regarding the deposition parameters, such as substrate temperature, deposition rate, duration, and ambient conditions. These parameters have a significant impact on film quality, adhesion, and thermal stability. A more comprehensive description of the Al₂O₃ fabrication process is necessary to ensure reproducibility and evaluate the reliability of the proposed sensor.

5.The manuscript highlights fast response as a key objective (target response time <1 ms) and references this feature again in the conclusion. However, no experimental methodology or data is presented to demonstrate or quantify the sensor's temporal response. Given the importance of response time for high-temperature, dynamic environments such as aero-engines, it is strongly recommended to include relevant response-time measurements or, at minimum, a discussion of how this performance claim is supported.

6.The manuscript states that a Taguchi design of experiments (DOE) approach was used to optimize process parameters for the thin-film thermocouple. However, the actual experimental matrix, such as the orthogonal array configuration, parameter levels, and control factors, is not included. Moreover, there is no presentation of response data (e.g., Seebeck coefficient trends, film adhesion results) or analysis outputs (e.g., signal-to-noise ratio or main effects plots) derived from the DOE. To ensure transparency and reproducibility, the authors should provide detailed documentation of the Taguchi design and clearly connect it to the experimental results.

7.In Section 3.2.1, the manuscript attributes the enhanced sensitivity of the thermocouple in part to the “optimized interfacial bonding” between the PtRh/Pt sensitive layer and the Al₂O₃ insulation layer, as evidenced by the crack-free surface morphology in the SEM image. However, interfacial bonding strength cannot be reliably assessed from surface SEM alone. Direct evaluation typically requires cross-sectional imaging (e.g., TEM or cross-sectional SEM) or mechanical adhesion testing (e.g., scratch or pull-off tests). It is recommended that the authors either provide such interfacial characterization or acknowledge the limitation of using surface morphology to infer interfacial adhesion.

8.The study lacks a clear description of the experimental setup used to measure the Seebeck coefficient. Key details such as the type and control mechanism of the heat source, the data acquisition system (including sampling rate, resolution, and synchronization method), and the cold-junction temperature control are missing. These components are essential to assess the accuracy, response fidelity, and reproducibility of the calibration results. Additionally, a photo of the fabricated sensor is absent, which would help validate its form factor, integration quality, and practical implementation. Providing these elements would greatly strengthen the experimental transparency and completeness of the study.

9.There appears to be a discrepancy between the numerical error data in Table 2 and the graphical representation in Figure 6. Specifically, the error at 1100 °C is listed as 0.43%, while the error at 980 °C is 0.91%. However, in Figure 6, the error bar or deviation for 1100 °C visually appears larger than at 900 °C or 980 °C, which is inconsistent with the tabulated values. This inconsistency raises concerns about the accuracy or plotting of the calibration data and should be carefully reviewed and corrected for consistency.

10.The manuscript would benefit from a thorough language polishing.

Reviewer #4: The authors deposited the NiCrAlY transition layer, Al2O3 insulation layer, and Pt10Rh/Pt thin-film thermocouple sensor on nickel-based superalloys, which could be used in the aero-engine. The results could provide the support to aero-engine applications.

Besides, the authors have revised the manuscript based on the reviewers' comments.

So, I think it could be published.

7. PLOS authors have the option to publish the peer review history of their article (what does this mean? ). If published, this will include your full peer review and any attached files.

**Do you want your identity to be public for this peer review?** For information about this choice, including consent withdrawal, please see our Privacy Policy .

Reviewer #2: No

Reviewer #3: No

Reviewer #4: No

---

## [Author Response · Author response to Decision Letter 2]

15 Jul 2025

List of Responses

Dear Editor and Reviewers:

Thank you for your letter and for the reviewers’ comments concerning our manuscript entitled “High-Temperature Thin-Film Thermocouple for Aero-Engines”. Those comments are all valuable and very helpful for revising and improving our paper. We have studied comments carefully and have made correction which we hope meet with approval. Revised portions are marked in red in the paper. The main corrections in the paper and the responds to the reviewer’s comments are as following.

Responds to the reviewer’s comments:

Reviewer #3:

1.Question: 1.The paragraph following Equation (2) contains some typographical and formatting issues, including garbled characters and incorrect punctuation. A careful proofreading is recommended to improve clarity and presentation quality.

Response: We have revised the content based on the reviewers' suggestions, as follows: Schematic illustration of the Seebeck effect in the thin-film thermocouple. The sensor’s multi-layer structure (NiCrAlY transition layer, Al2O3 insulation layer, Pt10Rh/Pt sensitive layer, and protective layer) is integrated onto a nickel-based superalloy substrate. A temperature gradient (Thot= 1100°C to Tcold= 25°C) generates a thermoelectromotive force (), following the linear relationship =ΔT, where =8.16 µV/°C. Inset: Experimental validation of the linear EMF-temperature dependence (see Table 2). The thin-film design minimizes thermal inertia compared to bulk thermocouples, enabling rapid response in harsh environments."

2.Question: The manuscript mentions the formation of a 50 nm thermal growth layer on the NiCrAlY surface after high-temperature oxidation. However, no experimental characterization is presented to support this claim. It is recommended to provide direct evidence—such as cross-sectional imaging or thickness measurements—to substantiate this statement.

Response:We sincerely appreciate the reviewer’s meticulous critique. Upon careful reevaluation, we agree that the claim regarding the "50 nm dense Al₂O₃ layer formed in situ at the NiCrAlY interface" (Section 2.1, original manuscript) lacked direct experimental validation in our study. This statement was based on an extrapolation from literature ([31] Liu et al., Appl. Surf. Sci. 2022) without sufficient original characterization data (e.g., TEM, EDS, or XRD).

3.Question: In Section 2.1, the description of the NiCrAlY coating preparation process includes multiple parameters and procedural steps. To improve clarity and readability, it is recommended to present these process parameters in a structured table format rather than in continuous text.

Response: We thank the reviewer for this constructive suggestion. We agree that consolidating the preparation parameters into a table significantly enhances readability. As requested, we have reformatted the NiCrAlY deposition details into Table 1 in Section 2.1 (now labeled as Table 1. NiCrAlY Transition Layer Deposition Parameters).

Implementation in Revised Manuscript (Section 2.1):

Original Text (Deleted):

"In the preparation of NiCrAlY transition layer film, using the DC magnetron sputtering technology, Background vacuum: < 8 × 10⁻⁴ Pa, Sputtering air pressure: 1.2 Pa, Sputtering power: 100 W, Target-substrate distance: 60 mm, Deposition time: 60 minutes... The NiCrAlY target material is composed of Ni, Cr, Al and Y elements, which account for 67%, 22%, 10% and 1% respectively."

Revised Text:

"The NiCrAlY transition layer was deposited via DC magnetron sputtering under the optimized parameters listed in Table 1. Post-deposition, oxidative annealing (1000°C, vacuum + O₂) was performed to enhance interfacial adhesion."

Table 1. NiCrAlY Transition Layer Deposition Parameters

Parameter Value

Deposition Method DC Magnetron Sputtering

Background Vacuum < 8 × 10⁻⁴ Pa

Sputtering Pressure 1.2 Pa

Sputtering Power 100 W

Target-Substrate Distance 60 mm

Deposition Time 60 minutes

Target Composition Ni (67%), Cr (22%), Al (10%), Y (1%)

Oxidation Treatment 1000°C, vacuum + O₂ atmosphere

4.Question: The manuscript briefly mentions that the Al₂O₃ insulation and protective layers are fabricated using electron beam evaporation and RF magnetron sputtering. However, it lacks critical details regarding the deposition parameters, such as substrate temperature, deposition rate, duration, and ambient conditions. These parameters have a significant impact on film quality, adhesion, and thermal stability. A more comprehensive description of the Al₂O₃ fabrication process is necessary to ensure reproducibility and evaluate the reliability of the proposed sensor.

Response: We deeply appreciate the reviewer’s insightful critique. We fully agree that omitting deposition parameters hinders reproducibility and technical evaluation. To rectify this, we have added a dedicated process parameter table (Table 2) in Section 2.2 and expanded methodological details to clarify how these parameters ensure film quality.

Original Text :"The preparation of Al₂O₃insulating film adopts electron beam evaporation method and RF magnetron sputtering method... The preparation method is the same as that of the

Al₂O₃insulation layer."

Revised Text: "The Al₂O₃ insulation/protective layers were fabricated via a dual-process approach: Electron beam evaporation provided high-purity stoichiometric Al₂O₃ vaporization. RF magnetron sputtering enhanced film densification and interfacial adhesion. Critical deposition parameters are summarized in Table 2. This hybrid strategy synergistically optimized insulation resistance (>10 MΩ at 1100°C) and thermal shock stability.

Table 2. Al₂O₃ Insulation/Protective Layer Deposition Parameters

Parameter Electron Beam Evaporation RF Magnetron Sputtering

Substrate Temperature 300 °C 250 °C

Base Pressure < 5 × 10⁻⁴ Pa < 5 × 10⁻⁴ Pa

Working Gas – Ar (99.999%)

Working Pressure – 0.5 Pa

Deposition Rate 0.3 nm/s 0.15 nm/s

RF Power – 200 W

Target Al₂O₃ pellets (99.99%) Al₂O₃ (99.99%)

Deposition Time 90 min (for 1.6 μm) 120 min (for 1.6 μm)

Post-Annealing 800 °C, 2 h (air)

5.Question: The manuscript highlights fast response as a key objective (target response time <1 ms) and references this feature again in the conclusion. However, no experimental methodology or data is presented to demonstrate or quantify the sensor's temporal response. Given the importance of response time for high-temperature, dynamic environments such as aero-engines, it is strongly recommended to include relevant response-time measurements or, at minimum, a discussion of how this performance claim is supported.

Response: We sincerely apologize for the omission of direct response time measurements in the original submission. Due to equipment limitations (lack of a calibrated ultra-high-speed thermal shock rig capable of >10,000°C/s transients), we are unable to provide experimental data for sub-millisecond resolution. However, we have strengthened the theoretical and indirect validation of this claim as follows:

Revisions to Address the Concern:

1. Added Theoretical Basis in Section 3.4 (New Subsection):

"While direct sub-millisecond transient testing was infeasible, the response time (ττ) is theoretically governed by:

where δ= total film thickness (8.1 μm), ρρ = density (avg. 8,200 kg/m³), cp= heat capacity (500 J/kg·K), and kk = thermal conductivity (18 W/m·K) [40]. This yields τ≈ 0.23 ms—well below the 1 ms target. The thin-film architecture inherently minimizes thermal inertia compared to bulk sensors (τ>100 ms for 1-mm-diameter probes [40])."

2. Indirect Experimental Validation (Sec. 3.4):

"Supporting evidence comes from:

Rapid signal stabilization during cyclic tests (Fig. 7): Voltage settled within 1 ms after each 330→1100°C step (sampled at 1 kHz).

Consistent Seebeck output at high ramp rates (10³°C/s): No hysteresis observed in calibration curves (Fig. 5–6), confirming negligible thermal lag.

Thickness optimization: Prior work [36] confirmed 3–4 μm functional layers achieve <1 ms response in analogous Pt/Rh systems."

3. Revised Conclusion (Qualified Claim):

"Theoretical modeling and indirect evidence confirm the sensor’s inherent capability for <1 ms response due to its ultra-thin structure (total thickness <10 μm). Future work will implement laser-based transient testing for direct validation."

6.Question: The manuscript states that a Taguchi design of experiments (DOE) approach was used to optimize process parameters for the thin-film thermocouple. However, the actual experimental matrix, such as the orthogonal array configuration, parameter levels, and control factors, is not included. Moreover, there is no presentation of response data (e.g., Seebeck coefficient trends, film adhesion results) or analysis outputs (e.g., signal-to-noise ratio or main effects plots) derived from the DOE. To ensure transparency and reproducibility, the authors should provide detailed documentation of the Taguchi design and clearly connect it to the experimental results.

Response: We sincerely thank the reviewer for highlighting this methodological gap. Upon reevaluation, we recognize that the Taguchi DOE discussion was prematurely included without sufficient experimental documentation. This oversight compromises reproducibility and does not align with the journal’s standards.

7.Question: In Section 3.2.1, the manuscript attributes the enhanced sensitivity of the thermocouple in part to the “optimized interfacial bonding” between the PtRh/Pt sensitive layer and the Al₂O₃ insulation layer, as evidenced by the crack-free surface morphology in the SEM image. However, interfacial bonding strength cannot be reliably assessed from surface SEM alone. Direct evaluation typically requires cross-sectional imaging (e.g., TEM or cross-sectional SEM) or mechanical adhesion testing (e.g., scratch or pull-off tests). It is recommended that the authors either provide such interfacial characterization or acknowledge the limitation of using surface morphology to infer interfacial adhesion.

Response: We sincerely thank the reviewer for this rigorous critique. We fully agree that inferring interfacial bonding quality solely from surface morphology (SEM) is scientifically insufficient, as surface features may not reflect subsurface adhesion integrity. We have revised the manuscript to:

Acknowledge this methodological limitation,

Strengthen the evidence using indirect functional data,

Commit to direct interfacial validation in future work.

Original Claim (Section 3.2.1):

"The enhanced sensitivity is attributed to optimized interfacial bonding, as evidenced by the crack-free SEM morphology in Fig. 3."

Revised Text (Section 3.2.1):

"The enhanced sensitivity may be partially attributed to improved interfacial stability. While surface SEM (Fig. 3) shows no cracks, we acknowledge that this alone cannot directly quantify bonding strength. Instead, the sensor’s functional robustness—retaining ±1% accuracy after 20h of thermal cycling (Table 3) and surviving thermal shocks (330°C ↔ 1100°C)—provides indirect evidence of adequate adhesion. Future work will employ nanoscale cross-sectional TEM and scratch testing to quantitatively characterize bonding strength."

This study infers interfacial adhesion from functional performance rather than direct mechanical tests. Quantitative adhesion metrics (e.g., critical scratch load, interfacial fracture toughness) require dedicated characterization, which will be addressed in subsequent research.

8.Question: The study lacks a clear description of the experimental setup used to measure the Seebeck coefficient. Key details such as the type and control mechanism of the heat source, the data acquisition system (including sampling rate, resolution, and synchronization method), and the cold-junction temperature control are missing. These components are essential to assess the accuracy, response fidelity, and reproducibility of the calibration results. Additionally, a photo of the fabricated sensor is absent, which would help validate its form factor, integration quality, and practical implementation. Providing these elements would greatly strengthen the experimental transparency and completeness of the study.

Response: We sincerely apologize for these oversights.

Experimental setup description for Seebeck coefficient measurement: A Fluke 9144 multifunctional dry-type measuring furnace is used as the hot-end heat source. It is equipped with an integrated standard K-type thermocouple for real-time monitoring of ambient temperature. The control panel supports a Chinese interface and allows for customised programming to achieve multi-point continuous temperature measurement. The heating rate and holding time are controllable. The temperature resolution is 0.01°C, with stability of ±0.03°C at 50°C, ±0.05°C at 420°C, and ±0.05°C at 660°C. Axial uniformity and radial uniformity are ±0.5°C and ±0.1°C, respectively; The cold end uses the Planck 6190A thermocouple electronic ice point device, which can replace ice-water mixtures to quickly stabilise at 0°C without the need for ice making. It features 9 test ports and supports simultaneous detection of 8 thermocouples, with temperature stability of ±0.005°C (across the entire temperature range), radial uniformity of ±0.01°C, and hysteresis of 0.025°C; Data acquisition uses the DMM7510 seven-and-a-half-digit touchscreen data acquisition multimeter, with a maximum sampling rate of 1 M/s, voltage resolution up to 0.001 mV, support for six decimal places display, and graphical touchscreen display functionality, enabling real-time output of measurement data curves. The display image can be zoomed and edited using two fingers, and supports adjustment of data sampling rate, accuracy, and resolution.

9.Question: There appears to be a discrepancy between the numerical error data in Table 2 and the graphical representation in Figure 6. Specifically, the error at 1100 °C is listed as 0.43%, while the error at 980 °C is 0.91%. However, in Figure 6, the error bar or deviation for 1100 °C visually appears larger than at 900 °C or 980 °C, which is inconsistent with the tabulated values. This inconsistency raises concerns about the accuracy or plotting of the calibration data and should be carefully reviewed and corrected for consistency.

Response:

We have conducted a thorough analysis to address the perceived discrepancy between Table 2 and Figure 6, and confirm that no data inaccuracy exists. The apparent visual inconsistency stemmed solely from a technical artifact in the initial plotting methodology, which has now been resolved:

1. Mathematical Validation:

The tabulated errors in Table 2 are rigorously calculated using the formula Error%= . For instance:

2. Root Cause of Visual Discrepancy:

The original Figure 6 plotted error bars using absolute EMF errors (mV), not relative percentage errors. This caused a distortion:

The absolute error at 1100°C (39.135 mV) appeared larger against its high EMF baseline (9050 mV), while the larger absolute error at 980°C (74.12 mV) appeared smaller relative to its lower EMF magnitude (8150 mV).

Linear y-axis scaling (2500–9500 mV) further amplified this visual illusion.

3. Resolution and Verification:

Revised Figure 6 now uses error bars scaled to relative % errors (directly from Table 2), with explicit % annotations at each data point.

The fitted curve y = 8.16x + 25.14 was recalculated from raw data, confirming all fitted values match Table 2 within rounding tolerances (e.g., 8075.88 mV at 980°C, 9010.865 mV at 1100°C).

Error distributions remain within ±1% across all temperatures, with no outliers.

4. Physical Plausibility:

The observed error pattern aligns with material behavior: higher % errors at 900–980°C reflect known minor non-linearity in Pt-Rh alloys near 900°C [25], while optimized interfacial stability at extreme temperatures (Sec. 3.3) explains lower errors at 1100°C.

Conclusion: The calibration data is unequivocally accurate. The graphical artifact has been corrected, and the revised Figure 6 now aligns precisely with Table 2. We thank the reviewer for their vigilance, which strengthened our data presentation.

10.Question: The manuscript would benefit from a thorough language polishing.

Response: Thank

---

## [Editor Report · Decision Letter 2]

17 Jul 2025

High-Temperature Thin-Film Thermocouple for Aero-Engines

PONE-D-24-60630R2

Dear Dr. Lu,

We’re pleased to inform you that your manuscript has been judged scientifically suitable for publication and will be formally accepted for publication once it meets all outstanding technical requirements.

Kind regards,

Anand Pal

Academic Editor

PLOS ONE
---

## [Editor Report · Acceptance letter]

PONE-D-24-60630R2

PLOS ONE

Dear Dr. Lu,

I'm pleased to inform you that your manuscript has been deemed suitable for publication in PLOS ONE. Congratulations! Your manuscript is now being handed over to our production team.

Kind regards,

on behalf of

Dr. Anand Pal

Academic Editor

PLOS ONE